# Strategic Decision-Making for Multi-Period Fleet Transition Towards Zero-Emission: Preliminary Study

Bogusław Bieda [ID], Roger Książek [ID], Katarzyna Gdowska *[ID] and Antoni Korcyl [ID]

Faculty of Management, AGH University of Krakow, al. Mickiewicza 30, 30-059 Krakow, Poland; bogbieda@agh.edu.pl (B.B.); roger@agh.edu.pl (R.K.); korcyl@agh.edu.pl (A.K.)
* Correspondence: kgdowska@agh.edu.pl; Tel.: +48-1261-743-34

**Abstract:** Municipal Solid Waste Management (MSWM) struggles with significant policy and operational challenges, particularly concerning collection routes for recyclables and fleet composition. Within the European Union, phasing out traditional fuel-based vocational vehicles, like garbage trucks, in favor of zero-emission alternatives, is mandatory to achieve sustainable development objectives. This paper presents a preliminary study on the problem of multi-period fleet transition from combustive fuels towards more eco-friendly fueling types. Initially developed for energy sector, the MARKAL framework was used here to support the technological transition of the fleet. The mixed-integer program was formulated for the Fleet Transition Problem (FTP), a simplified theoretical problem. The objective of the FTP and a mixed-integer linear program used to solve it is minimizing the overall cost of fleet modernization throughout a multi-phase planning horizon so that the sustainable transition of the fleet can be assured. Computational experiments run on randomly generated data instances affirmed the model's effectiveness in strategizing fleet transition. This research outlines a multi-period model for transitioning to a zero-emission fleet and demonstrates the FTP's potential for strategic decision-making. Notably, the study observes consistent reductions in permissible emissions across the planning horizon.

**Keywords:** MARKAL; fleet optimization; solid waste management; mixed-integer linear programming; European Green Deal; optimization; recyclables collection; fleet sustainable transition





## 1. Introduction

The nexus between systemic circular economy principles and the efficient collection of solid waste is profoundly intricate. The realization of a systemic circular economy is contingent upon the existence of a robust and efficient solid waste collection system. Efficient and intelligent Municipal Solid Waste Management (MSWM) is indispensable for environmental cleanup, particularly when executed with technical precision and in alignment with local market dynamics and sustainability. Technological advances within the MSWM sector play a critical role in enhancing service efficiency and promoting environmental sustainability, ultimately benefiting local municipalities, solid waste collection companies, stakeholders, and citizens in their pursuit of an improved quality of life and safety [1,2]. Nevertheless, the multifaceted nature of municipal solid waste collection continues to pose a formidable challenge for decision-makers at both local and national levels. These difficulties cover various aspects, starting with educational initiatives focused on encouraging public adoption of the circular economy concept. Simultaneously, they extend to develop decision support systems customized to enhance both strategic and operational aspects of Municipal Solid Waste Management, including managing the solid waste collection fleet and routing solid waste collection vehicles.

The provision of essential municipal services and the alignment of their technologies with environmental mandates hold paramount significance for the regional, national, and global economies, particularly within the European Union context. Macroeconomic

factors intricately impact the effectiveness of predictive decision-making tools in waste management, warranting their integration into a comprehensive strategic analysis and decision support model. To address this, tools developed for pinpointing key facets of innovative technologies within a circular economy's macroeconomic landscape can be leveraged [3,4]. However, it is important to note that this paper's focus on a pilot study employing the MARKAL (MARKet ALlocation) framework for transitioning municipal service vehicle fleets primarily centered on emissions analysis and macroeconomic aspects were not its primary aim. Municipal Solid Waste Management is intrinsically linked with navigating a series of policy and governance challenges. At present, two pivotal concerns surface prominently: (1) separate recyclables collection routing, and (2) vocational fleet composition. The adoption of a separate recyclables collection approach streamlines the recycling process by providing recycling centers with pre-sorted recyclable materials. However, it necessitates repeated visits to the solid waste pickup points. These points are visited initially for collecting mixed household waste, and subsequently for retrieving segregated recyclables. The allocation of distinct garbage trucks dedicated to specific recyclable types amplifies the frequency of visits to each pickup point. The operational problem of the MSWM is intricately intertwined with routing problems, owing to the salient influence of solid waste collection vehicles' characteristics, such as size, range, and emitted noise, in this intricate process [5–9]. Consequently, an imperative arises to strategically optimize the fleet used for the solid waste collection service, so that multiple visits at the pickup points may not result in increased emissions produced by the vehicles used [10–12].

In this paper, bringing the results of a preliminary study, the main research question is how useful is an approach inspired by the MARKAL framework [13,14] for planning a gradual fleet transition toward zero-emission and sustainability minimizing the total cost for the transition in a multi-period planning horizon. This is why the authors intentionally reduced the number of factors to be taken into consideration in the decision-making; the problem was simplified to minimizing the total cost of replacement and modernization of the fleet and there was the requirement to decide once over the planning horizon about each vehicle to modernize it or to replace. The decision-making is far more complex as there are numerous economic, social, and environmental parameters of the population of a location to be served, moreover, there are more tools and strategies to be applied for more optimized waste management and treatment. Moreover, the choice of technologies to which the existing one can potentially be shifted needs further elaboration to include vehicles and devices that will be most demanded depending on the nature of the urban solid waste, whether it is a high-standard residential regions, commercial regions, mixed commercial and residential regions, industrial regions, etc.

The main contribution of the paper is the newly developed MARKAL-based mixed-integer linear programming model for the Fleet Transition Problem, which aims at finding optimal schedules for modernization and substitution vehicles used for solid waste collection services; the proposed model is an initial attempt to collect and structure information for making technical decisions regarding the reduction in greenhouse gas emissions. Obtained schedules are optimal which means that the total cost of modernization and substitution are minimal. In this study, the original MARKAL model utilizes mathematical modeling to evaluate and strategize technology shifts within energy systems, enabling scenario-driven analysis. By optimizing technology selections and investments, it targets sustainability goals, balancing costs, efficiencies, and environmental impacts. This method empowers decision-makers by offering valuable insights into the optimal pathways for transitioning to sustainable energy solutions. This paper is structured as follows: in Section 1.1, we briefly present the importance of the fleet transition problem in Solid Waste Management from the perspective of obligations mandated by the European Green Deal. In Section 2, we present the MARKAL framework and its applications as well as the newly developed Fleet Transition Problem together with an associated mixed-integer linear program. Next, in Section 3, we report computational experiments. In Section 4, we delineate potential avenues for future research.

### 1.1. Fleet Transition in Solid Waste Management

The European Green Deal obliges EU member countries to attain climate neutrality by 2050 [15]. The realization of a resource-efficient Europe hinges on the comprehensive and large-scale development and implementation of a systemic circular economy. To accomplish this, the adoption of urban and regional circular design solutions is imperative. Effective implementation requires a nuanced policy mix that maximizes synergies and addresses the inherent trade-offs across diverse domains and policy areas. As a result, it is essential to provide local authorities, citizens, and other stakeholders with a collaborative, science-informed decision-making environment conducive to exploring various waste and resource management alternatives. Such an environment should also enable the assessment of these alternatives' impacts on environmental resilience, spatial quality, and overall quality of life.

A profound comprehension of the circular economy concept mandates a comprehensive understanding of the intricate interplay between socioeconomic and environmental dynamics, as well as the physical built environment. This expanded comprehension renders the circular economy concept more practicable and pragmatic. Consequently, there exists a pressing need for innovation in several key areas, including the integration of dynamic resource flow modeling, resource allocation in conjunction with urban and regional planning and design, and the consideration of human behavioral factors. In the EU, there is a strong recommendation, even a requirement, to withdraw diesel-powered vocational vehicles and compose the fleet using zero-emission alternatives, e.g., hybrid, hydrogen, or electric cars [16,17]. In Poland, where electromobility still has a small share of the market [18], the Act on Electromobility and Alternative Fuels required organizations responsible for delivering services within a municipality to undertake actions towards transforming their fleet compositions, so that by 2028, 30% of their vehicle can be considered as zero-emission vehicles (ZEV). To achieve this goal, specific interim targets have been established: 5% by 2021, 10% by 2022, and 20% by 2025 [19]. While emerging propulsion technologies and electromobility enhance the environmental efficiency of the transportation sector, the prevalence of older, less eco-friendly vehicles in car fleets undermines these advancements due to increased resource consumption and environmental degradation caused by extensive servicing and parts replacement [20].

However, the integration of electric vehicles (particularly electric solid waste collection vehicles) into municipal fleets presents significant challenges due to disparities in key characteristics compared to their conventional counterparts. Differences in range, capacity and noise level necessitate adaptations in fleet management strategies [21–23]. Notably, the limited range of electric trucks may necessitate rerouting efforts. In contrast, the reduced noise level allows for nocturnal solid waste collection. As the proportion of electric garbage trucks in municipal fleets continues to grow, the demand for effective tools to manage a heterogeneous fleet becomes increasingly pressing. Amidst evolving waste management laws, there is a pressing need for effective decision support systems. These systems are imperative for ensuring seamless integration of electric vehicles, optimizing routes, and enhancing the overall efficiency of municipal solid waste collection operations. It should not be overlooked that strategic decisions to replace the fleet with differently powered vehicles can significantly affect the operational management of transport tasks, which is very visible and acute in the case of urban public transport [24].

Managers of diverse companies and service-providing agencies reliant on transportation confront the intricate challenge of fleet replacement to meet environmental mandates. This challenge is further intensified by the necessity to reconcile service quality, which may result in emissions, with the economic optimization of in-service vehicles, aligning them with the company's operational efficiency and its long-term strategy of transitioning the fleet towards a more environmentally sustainable composition. Striking a harmonious balance to establish an optimal scenario for the gradual, multi-period fleet replacement with more environmentally friendly options is a complex undertaking [25]. Making informed—if not ideal—decisions in this context is a formidable task, as decision-makers must rely on

economic and ecological analyses as well as their expertise. Given the intricacy and multi-period nature of this issue, the utilization of decision-support tools becomes indispensable.

Such a problem is faced in maritime transportation—where before creating an optimization model, informed market analysis was conducted—to identify the minimal number of electric vessels needed to fully replace the current diesel fleet, while considering multiple factors, including transport demand, vessel quantity, and environmental sustainability [26]. The study by Ahani et al. [27] presents an innovative framework designed to support urban freight transportation operators in optimizing their fleet configuration to minimize overall costs while adhering to regulations related to vehicle size and type and complying with specific city zones defined by local authorities. The framework takes into account various cost factors, including vehicle acquisition, energy consumption, emissions, maintenance, salvage, and labor across different vehicle types. The outcome of the model provides insights into the optimal fleet composition, specifying the necessary quantity, size, and types of vehicles required for operation in different city areas throughout the planning period.

Maritime transportation faces a similar challenge. In a recent study by Prina et al. [26], market analysis preceded the development of an optimization model. This aimed to determine the minimum number of electric vessels necessary to replace the current ON-powered fleet, considering factors like transport demand and environmental sustainability. Another study by Ahani et al. [27] introduced a novel framework for urban freight transportation operators. This framework optimizes a fleet composed of various types of vehicles, considering the demand for services, geographical and demographic characteristics of the city, and legal obligations along with costs of purchase, refueling, maintenance, and operations as well as emission. The model provides insights into the ideal fleet composition, specifying vehicle quantity, size, and types required for serving various cities in a planning horizon.

In the research to which this paper refers, the MARKAL framework which for 40 years is applied in the energy sector for generating schedules for transition technologies from the currently used ones to the newer or more innovative ones was found interesting in terms of mixed-integer linear programming models used for planning the multi-period technology transition. The MILP model which is the core of the MARKAL framework was adjusted for MSWM needs in the area of technology transition. Neither the question of energy transition associated with solid waste management and how these segments are connected through tools and methods based on the principles of the circular economy nor how to deal with the energy transition established by European Union member countries was included in this research. Deep technical analysis and parameterization of available technologies should be performed by experts at the stage of system analysis before the optimization stage, so the numeric results can be used as input data for the MILP model and support the informed decision-making process on updating the fleet.

## 2. Materials and Methods

In this paper, we present a preliminary study on potential utilization of the MARKAL-based approach to the strategic decision-making problem of gradual fleet transition from one type of fuel to another. The main research question was to assess the usefulness of the MARKAL framework for strategic decision making influenced by a small number of decision criteria. The strategic problem of municipal solid waste management concerning the gradual modernization and replacement of the fleet as an optimization problem where the total cost of replacing and modernizing the fleet of waste collection vehicles is minimized, which means that only 2 criteria were considered: the total cost of replacement and modernization which was minimized and the range of introduced transition which covered the entire fleet, i.e., on each vehicle the decision to replace or to modify had to be made. At the same time, the constraints must be satisfied, i.e., the gradual increase in the limitation on the total available emissions limit over a planning horizon. We formulated the Fleet Transition Problem (FTP) and propounded to solve it using a MARKAL-based mixed-integer linear program.

### 2.1. MARKAL—An Overview

The MARKAL (MARKal ALlocation) model was initially formulated as a linear program to select emerging energy technologies to fulfill best the needs of a national energy system in a chosen set of regions. It evolved towards a technology-rich framework capable of predicting energy trends over multiple periods. The model facilitates decisions regarding equipment investments, operations, and regional primary energy supply. In scenario analysis including alterations in demand for energy, MARKAL determines whether to optimize existing infrastructure or invest in new technologies. The assessment of technologies under consideration takes into account the economic aspects of primary energy supply. In essence, MARKAL helps to establish energy equilibrium across technologies of different technological levels to minimize overall global energy expenses [13,14].

MARKAL stands as an all-encompassing framework delving into the entirety of the energy system, calculating an inter-temporal partial equilibrium within the energy markets. Essentially, MARKAL-based models aim at balancing quantities and prices of fuels, i.e., determining optimal prices for energy aligned with the demanded quantities sought by consumers which can be fulfilled by suppliers. Moreover, all the investments needed to reach that balance by the end of the planning horizon are optimally distributed across the planning period, so that the total surplus is maximized. The MARKAL-based framework was widely utilized to assess developmental directions and the velocity of technological transition mainly for the energy sector, also towards its decarbonization, in Canada [28,29], Switzerland [30], Malaysia [31], Russia [32], the UK [33–36], the USA [37,38], Austria [39], China [40–43], Portugal [44–46], Italy [47], Iran [48], Turkey [49], Kazakhstan [50], Ireland [51], Poland [52], Greece [53], and Bulgaria [39].

In addition, Salvia et al. [54–56], using the MARKAL framework, developed a detailed model to analyze the human–system interactions in the Basilicata region, Italy, focusing on solid waste management. This aimed to conform to Italian regulations, ensuring an efficient regional solid waste management plan. They conducted a sensitivity analysis, particularly examining the influence of landfill fees on solid waste processing decisions. The study aimed to create a sustainable, cost-effective, and efficient waste management strategy, expanding the MARKAL model's applications beyond the energy sectors. Key factors affecting the integrated system include landfill volume restrictions, greenhouse gas emissions, and solid waste disposal charges. However, the success of such a plan depends significantly on human factors, emphasizing the importance of training, organizational strategies, and public engagement for effective implementation. The limitation of this study does not refer to the solid waste collection vehicles separately but just as a part of the system.

### 2.2. Fleet Transition Problem

Let set $T$ denote the planning horizon which consists of $t$ planning periods, while set $R$ denotes the types of fueling available for the solid waste collection vehicles. At each planning horizon $t$, the demand for the total number of vehicles to fulfill the solid waste collection service is known. We also know in advance two types of costs: (1) costs $c_{r,l}$ for modernizing a vehicle, so that it is shifted from operating on fuel $r$ to fuel $l$, and (2) costs $C_r$ that must be incurred to purchase a new vehicle operating on the fuel $r$. Let $b_t$ denote the total allowable emissions that moving solid waste collection vehicles can generate in each period $t$. For each vehicle fuelled with fuel $r$, the amount of emissions generated ($e_r$) is known. We also know in advance $I_r$ the initial number of vehicles fueled with fuel $r$ in the possession of a solid waste collection company.

The decision criterion is the minimum total cost of fleet modernization and replacement discounted over time according to an assumed discount rate of $q$. The costs incurred are related to the decisions made about (1) the number of vehicles shifted from fuel $r$ to fuel $l$ (denoted with decision variable ($f_{r,l,t}$) and (2) the number of purchased new vehicles fuelled with greener fuel $r$ (denoted with decision variable ($u_{r,t}$). The decision variable $x_{r,t}$ stands for the number of vehicles fuelled with fuel $r$ in the possession of the company

in planning period $t$. The notation used in the formulas of the MILP model for FTP is presented in Table 1.

**Table 1.** The notation used in the MILP model for FTP.

| **Sets** | |
|---|---|
| $T$ | – planning horizon, i.e., the set of planning periods |
| $R$ | – the set of the types of fueling available for the solid waste collection vehicles |
| **Parameters** | |
| $d_t$ | – the number of solid waste collection vehicles needed in period $t$ to perform solid waste collection service |
| $c_{r,l}$ | – the cost of shifting a vehicle from fuel $r$ to fuel $l$ |
| $C_r$ | – the cost of purchasing a vehicle fuelled with fuel $r$ |
| $b_t$ | – maximum allowable emission in period $t$ |
| $e_r$ | – emission produced by a vehicle operating on fuel $r$ per its capacity unit |
| $q$ | – discount rate |
| $I_r$ | – the initial number of vehicles operating on fuel $r$ possessed by the company |
| **Decision variables** | |
| $f_{r,l,t}$ | – the number of vehicles operating on fuel $r$ shifted to fuel $l$ in the planning period $t$ |
| $x_{r,t}$ | – the number of vehicles operating on fuel $r$ possessed by the company in the planning period $t$ |
| $u_{r,t}$ | – the number of vehicles operating on fuel $r$ purchased by the company in the planning period $t$ |

A mixed-integer linear program was formulated to address the Fleet Transition Problem employing Formulas (1)–(11).

$$\min: \sum_{t \in T} \left( \sum_{r \in R} \sum_{l \in R} c_{r,l} * f_{r,l,t} + \sum_{r \in R} C_r \cdot u_{r,t} \right) \cdot q^t; \tag{1}$$

$$\sum_{k \in R} f_{r,l,0} = I_r, r \in R; \tag{2}$$

$$\sum_{r \in R} x_{r,t} = d_t, t \in T; \tag{3}$$

$$\sum_{r \in R} e_r \cdot x_{r,t} \le b_t, t \in T; \tag{4}$$

$$x_{r,t} = \sum_{l \in R} f_{l,r,t} + u_{r,t}, r \in R, t \in T : t > 0; \tag{5}$$

$$f_{r,r,t} \le u_{r,t}, r \in R, t \in T; \tag{6}$$

$$\sum_{l \in R} f_{r,l,t} = \sum_{l \in R} f_{l,r,t-1} + u_{r,t-1}, r \in R, t \in T : t > 0; \tag{7}$$

$$\sum_{r \in R} u_{r,t} = d_t - d_{t-1}, t \in T : t > 0; \tag{8}$$

$$f_{r,l,t} \ge 0, f_{r,l,t} \in C, r \in R, l \in R, t \in T; \tag{9}$$

$$x_{r,t} \ge 0, x_{r,t} \in C, r \in R, t \in T; \tag{10}$$

$$u_{r,t} \ge 0, u_{r,t} \in C, r \in R, t \in T; \tag{11}$$

The objective function (1) aims at minimizing the total cost associated with modernizing existing vehicles (i.e., shifting them from fuel $r$ to $l$) and purchasing new ones. Constraint (2) forces the search for a solution for the initial composition of the fleet possessed by the solid waste collection company. The required number of vehicles $d_t$ in each planning period $t$ is ensured by constraint (3). Constraint (4) ensures that the allowable emission limit $b_t$ is maintained for each period $t$. Accordingly, Constraints (5)–(7) and Constraint (8), respectively, allow the modernization of a given number of vehicles and (or) the purchase of new ones, by maintaining the correct balance for the fleet composition in subsequent planning periods during the planning horizon.

### 3. Computational Experiments—Results and Discussion

As a validation test, the following optimization problem has been formulated: Municipal Solid Waste Management (MSWM) needs to plan and manage the modernization of their vehicle and the purchase of new solid waste collection vehicles so that in the next years the requirements for reducing the permissible emission standards for used vehicles can be met. To keep the example simple, let us assume that the solid waste collection vehicles under consideration can run on four types of fuel: $A$, $B$, $C$, and $D$. We consider fuel $A$ as the most popular currently but causing significant environmental impact, while fueling vehicles with $B$, $C$, and $D$ is expected to be more eco-friendly. We neither consider here technological details of each fueling system nor their efficiency in different weather conditions throughout the year. Moreover, as the problem is simplified it cannot estimate the effects caused by the transition in terms of new energy and fleet maintenance costs, i.e., the costs involving the maintenance and/or replacement of these electric motors, as well as the possible costs of treatment and final disposal.

Note that the example presented in this section is just an illustration to show what outputs can we obtain, as it is a preliminary study, the computations were conducted for randomly generated data instances which have no relation to currency or technical parameters. As the intention was to examine the potential usefulness of the MARKAL framework for fleet transition, the presented results are an illustration. They are based on randomly generated data instances that do not correspond with any specific fueling type. Not to confuse the readers, we replaced the names of technologies with letters $A-D$, so we can concentrate on the FTP model and not on a specific case.

As the MSWM continues to grow, we know the projected demand for the number of vehicles which from the current 50 vehicles should increase to 110 vehicles in a 5-year planning horizon. The projected demand for vehicles is shown in Figure 1.

The MSWM knows the number of vehicles currently owned, the cost of purchasing a vehicle powered by each fuel type, and the total annual emissions of each type of vehicle. The data is presented in Table 2.

**Table 2.** Values of the parameters characterizing the chosen method of vehicle power supply adopted in the computational example.

| Fuel | The Number of Vehicles Possessed | The Cost of Purchasing a New Vehicle [Normalized] | Annual Emission [Normalized] |
|------|------|------|------|
| $A$ | 50 | 30 | 2.0 |
| $B$ | 0 | 35 | 1.0 |
| $C$ | 0 | 45 | 0.5 |
| $D$ | 0 | 50 | 0.0 |

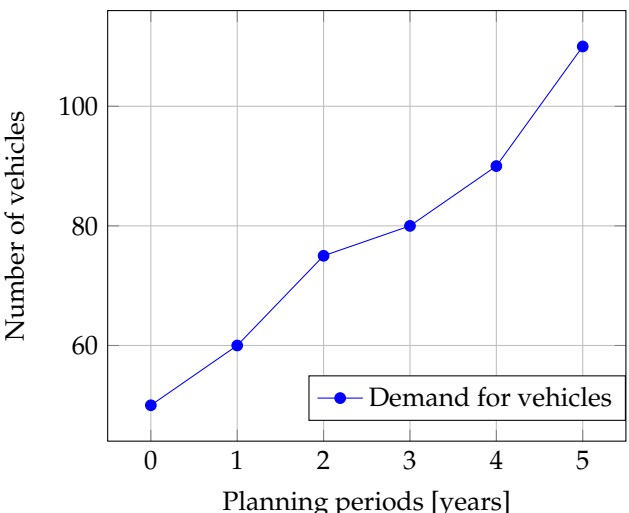

**Figure 1.** Forecasted demand for vehicles in subsequent planning periods.

In addition, the MSWM knows the cost of modernization of a vehicle from one fuel to another which results in reduced emissions. The modernization costs are shown in Table 3.

**Table 3.** The normalized cost of modernization of a vehicle from one fuel to another.

|   | **A** | **B** | **C** | **D** |
|---|---|---|---|---|
| *A* | – | 3 | 10 | 15 |
| *B* | – | – | 10 | 15 |
| *C* | – | – | – | 20 |
| *D* | – | – | 15 | – |

Computational experiments were undertaken to find the solution for the exemplary FTP instances presented above. The provided MIP model could effectively solve using a standard computing setup, in line with typical computational resources available for use within the company. The solver employed for this analysis was GUROBI 9.0.1 [57], run on a computer equipped with a dual-core Intel Core i7-4710HQ CPU operating at 2.50 GHz and 16 GB of RAM.

Throughout the planning horizon, there is a notable decline observed in the maximum permissible emissions. Commencing from the current emission levels (i.e., 100 units per year), there is a discernible reduction over successive planning periods, reaching the minimum allowable annual emissions of five units per year in the latest planning period. The diminishing trend of the annual emission is graphically depicted in Figure 2.

In the instance under examination, the optimal solution was found, which minimizes the total cost for modernization and replacement of the fleet during the predefined planning horizon, considering an annual discount rate of 7%. The computed optimal total cost was 163.67 units. The MWSM under examination possessed 50 vehicles in the initial planning period. Conversely, in the ultimate planning period, as dictated by the vehicle demand data, a total of 110 vehicles are necessary. Consequently, within this planning horizon, there was a necessity to procure 60 new vehicles, including both purchasing 60 units and modernizing 50 of them. It should be highlighted that the FTP model newly developed in this paper does not account for the selling or disposal of any spare parts of the existing vehicle fleet. Table 4 meticulously outlines the comprehensive schedule detailing the planned acquisitions for each vehicle category during each planning period. In total, the stipulated acquisitions amounted to 60 vehicles, the predominant count among them being 47 *D*-powered vehicles.

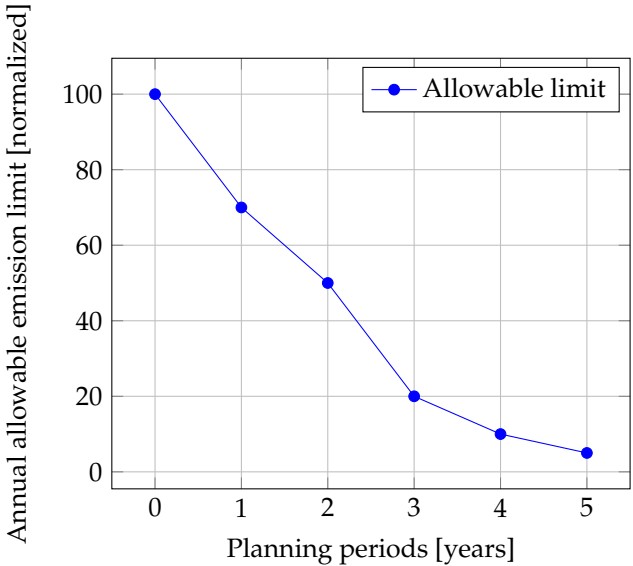

**Figure 2.** Total allowable emission limits for the planning horizon.

**Table 4.** The number of vehicles operating on each fuel purchased by the company in each planning period.

| Planning Periods [Years] | 0 | 1 | 2 | 3 | 4 | 5 |
|:---:|:---:|:---:|:---:|:---:|:---:|:---:|
| *A* | – | – | – | – | – | – |
| *B* | – | 10 | 2 | – | – | – |
| *C* | – | – | 1 | – | – | – |
| *D* | – | – | 12 | 5 | 10 | 20 |

Figure 3 shows the number of available vehicles fueled by each type of fuel over each planning horizon. Due to the decreasing emission limit over the planning horizon, emission-fueled vehicles are gradually being replaced by emission-free vehicles. In the last planning horizon, the vehicle fleet consists of four *B*-fueled vehicles, two *C*-fueled vehicles, and 104 *D*-fueled vehicles.

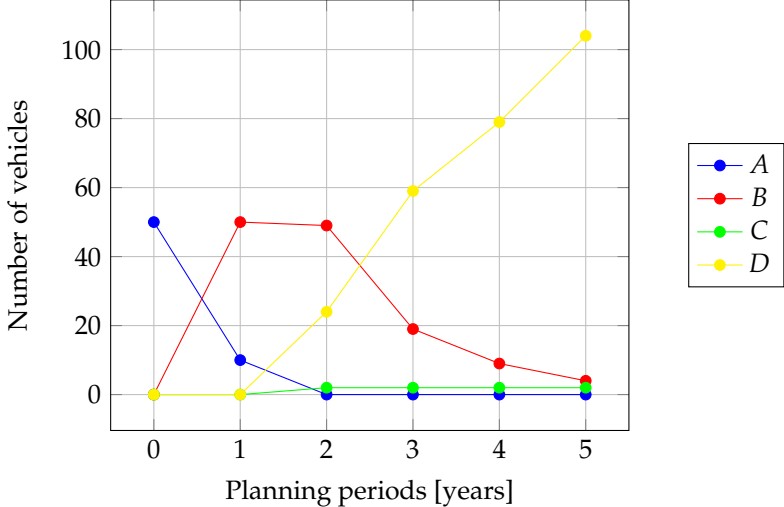

**Figure 3.** The number of vehicles operating on each fuel possessed by the company in each planning period.

Figure 4 shows the computed values obtained of flow variable $f_{r,k,t}$, denoting the count of vehicles transitioning from fuel type $r$ to fuel $k$ in each period. In the initial planning period $t = 0$, 40 vehicles running on *A* were modernized to run on *B*, while 10 vehicles

remained unchanged. However, in the subsequent period $t = 1$, all vehicles powered by fuel $A$ were upgraded to be $D$-fueled, 37 remained unaltered, while three vehicles underwent modifications: one transformed into a $C$-fueled vehicle, and two were modified into $D$-fuel-type vehicles. Modernization of the vehicle fleet has always been carried out in the direction of lowering emissions. In the final planning phase, no further alterations occurred, with only 20 $D$-fueled vehicles procured.

$t = 0$

|   | $A$ | $B$ | $C$ | $D$ |
|---|---|---|---|---|
| $A$ | 10 | 40 | 0 | 0 |
| $B$ | 0 | 0 | 0 | 0 |
| $C$ | 0 | 0 | 0 | 0 |
| $D$ | 0 | 0 | 0 | 0 |

$t = 1$

|   | $A$ | $B$ | $C$ | $D$ |
|---|---|---|---|---|
| $A$ | 0 | 0 | 0 | 10 |
| $B$ | 0 | 37 | 1 | 2 |
| $C$ | 0 | 0 | 0 | 0 |
| $D$ | 0 | 0 | 20 | 0 |

$t = 2$

|   | $A$ | $B$ | $C$ | $D$ |
|---|---|---|---|---|
| $A$ | 0 | 0 | 0 | 0 |
| $B$ | 0 | 17 | 0 | 30 |
| $C$ | 0 | 0 | 1 | 0 |
| $D$ | 0 | 0 | 0 | 12 |

$t = 3$

|   | $A$ | $B$ | $C$ | $D$ |
|---|---|---|---|---|
| $A$ | 0 | 0 | 0 | 0 |
| $B$ | 0 | 9 | 0 | 10 |
| $C$ | 0 | 0 | 2 | 0 |
| $D$ | 0 | 0 | 0 | 54 |

$t = 4$

|   | $A$ | $B$ | $C$ | $D$ |
|---|---|---|---|---|
| $A$ | 0 | 0 | 0 | 0 |
| $B$ | 0 | 4 | 0 | 5 |
| $C$ | 0 | 0 | 2 | 0 |
| $D$ | 0 | 0 | 0 | 69 |

$t = 5$

|   | $A$ | $B$ | $C$ | $D$ |
|---|---|---|---|---|
| $A$ | 0 | 0 | 0 | 0 |
| $B$ | 0 | 4 | 0 | 0 |
| $C$ | 0 | 0 | 2 | 0 |
| $D$ | 0 | 0 | 0 | 84 |

**Figure 4.** The number of vehicles shifted from fuel $r$ to fuel $l$ in each planning period $t$ [year].

It is worth noting that the Fleet Transition Problem introduced in this study has been effectively validated through the resolution of illustrative cases. However, future research endeavors should aim to assess the FTP's performance using historical data derived from an actual solid waste collection company. Given the prevailing recommendation for the incremental modernization of solid waste collection fleets in this domain, this investigation has demonstrated that the MARKAL-based model, originally designed for the energy sector, serves as a pertinent and valuable tool for the transportation sector, particularly in the context of strategic decisions concerning substantial fleet investments. Subsequently, forthcoming research endeavors should incorporate similar studies, employing the FTP on datasets derived from actual operations, and delving into more comprehensive analyses of the determinants influencing fleet replacement and modernization decisions.

Note that the FTP is not a prediction model. It is an optimization model—a mixed-integer programming model. So, using the FTP we cannot estimate the effects caused by the transition in terms of new energy and fleet maintenance costs. Moreover, costs involving the maintenance and/or replacement of these electric motors, as well as the possible costs of treatment and final disposal are not included directly in this model; they may be included indirectly in the cost of modernization or replacement, but must be computed separately before the data instance for FTP is prepared. For such input data, long-term costs are considered in this particular MILP model. If there is a need to incorporate these costs directly in a long-term exploitation period, a new model must be developed.

By integrating the budget issue into the MARKAL-based model for the fleet transition problem, a more comprehensive and nuanced decision support tool will be developed. This will empower stakeholders and decision-makers with the tools necessary to navigate the complex landscape of fleet optimization, considering both economic constraints and

environmental imperatives. It will be necessary to investigate the best way to make meeting these goals economically viable, especially with the next few years of greater impacts from inflation and gradual slowdown in the global economy as a whole, especially in member countries of the European Union. Furthermore, the MARKAL-based approach seems to be potentially useful for optimizing the proportion of modified and purchased low-emission vehicles within the overall fleet over a planning horizon obtained in the available budget. This can be achieved by imposing a predetermined percent for specific periods, aligning with demand for solid waste collection services, budget limitation for fleet transition, and environmental targets. This addition to the model will allow for a more nuanced exploration of scenarios that actively contribute to meeting regulatory conditions while balancing economic considerations.

## 4. Conclusions

It is imperative to emphasize again that this paper represents preliminary investigations into the potential applicability of the MARKAL model in strategizing the transition of fleets within a solid waste collection enterprise. We have presented a MARKAL-inspired MILP model for a multi-period project of modernization and replacement vehicles toward having a zero-emission fleet. The example of the Fleet Transition Problem for which computational experiments were conducted illustrates a potential application of the developed FTP model for strategic decision-making.

In future research, the emphasis should be placed on incorporating the financial aspect to the decision-making process regarding the transition of the solid waste collection fleet through modification and gradual replacement of the vehicles. Two directions of research can be distinguished in this context. Firstly, a scenario where a fixed budget is allocated for each planning period. In this case, the model will need to optimize the mix of modernization and replacement costs, determining how to allocate the budget for vehicle modernization and replacement while adhering to the specified constraints. This approach will provide valuable insights into the optimal allocation of financial resources, balancing the integration of modernized and newly bought vehicles. Secondly, an approach involving an ideal situation with an unlimited budget. This entails a comprehensive examination of the cost implications associated with technology changes, specifically fuel-type modifications. This budget-independent analysis aims to uncover the economic considerations intrinsic to technology transitions. It will result in obtaining the benchmark solution where the emission is undoubtedly minimized.

**Author Contributions:** Conceptualization, A.K. and R.K.; methodology, K.G.; software, R.K.; validation, R.K., A.K. and K.G.; formal analysis, B.B.; investigation, R.K.; resources, A.K.; data curation, R.K.; writing—original draft preparation, K.G. and R.K.; writing—review and editing, A.K., B.B., R.K. and K.G.; visualization, R.K. and K.G.; supervision, B.B. and A.K.; project administration, K.G.; funding acquisition, A.K. All authors have read and agreed to the published version of the manuscript.

**Funding:** This work was supported by the Program "Excellence Initiative—Research University" with the AGH University of Krakow, Poland.

**Institutional Review Board Statement:** Not applicable.

**Informed Consent Statement:** Not applicable.

**Data Availability Statement:** No new data were created or analyzed in this study. Data sharing is not applicable to this article.

**Conflicts of Interest:** The authors declare no conflict of interest.

**Abbreviations**

The following abbreviations are used in this manuscript:

FTP     Fleet Transition Problem
MILP    Mixed-Integer Linear Programming
MSWM    Municipal Solid Waste Management
OR      Operations Research
SWM     Solid Waste Management
ZEV     Zero-Emission Vehicle

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
