# Peer review of "Strategic Decision-Making for Multi-Period Fleet Transition Towards Zero-Emission: Preliminary Study"

_sustainability, doi:10.3390/su152416690_

Round 1

Reviewer 1 Report

Comments and Suggestions for Authors

Dear Authors, kindly consider the comments marked on the attached file, and revise the manuscript accordingly.

Comments on the Quality of English Language

Minor grammatical corrections are required.

Author Response

The authors are  immensely thankful to the reviewers for their meticulous assessment and constructive criticism, contributing significantly to the refinement and strengthening of our research findings in the paper. Below we address the Reviewers’ comments one by one, changes in the manuscript are marked in pink.

Reviewer 2 Report

Comments and Suggestions for Authors

This work applied MARKAL model to solve practical and important Fleet Transition Problems (FTP) in the Municipal Solid Waste Management (MSWM). Overall, the work is well designed and written, but there are some issues the authors need to address:

1) For all the figures and tables, please add units in them (e.g., year for x-axis. For costs, if it is normalized, please denote it is normalized, otherwise, please add units in dollar or euro.

2) To remain consistency with the description in the context, it is better to allocate Figure 1 ahead of Table 2, put Table 4 ahead of Figure 3 

3) For the model, please explain why two q with different meanings in Table 1. Why introduce a totally unknow symbol k in your Eq 1a, 1b, 1e, 1g? Are they just typos? do you mean "l" or fuel "l"? Only "l" mentioned in Line 203, page 5. Please avoid carelessness in mathematical model, otherwise it will make your readers question and doubt your ability and all your results in the manuscript.

4) grammar and typo: Line 150, Page 4, should be "At the same time". Line 175-176, Quebec is part of Canada, right? why repeat China twice? Line 219, Page 6, should be "computational experiments". Please be careful.

Comments on the Quality of English Language

Except minor typos, the manuscript is well written. 

Author Response

(The authors gave the same response as above.)

Reviewer 3 Report

Comments and Suggestions for Authors

GENERAL COMMENTS

- The manuscript meets the minimum requirements to be published in the Sustainability Journal. However, it is necessary to adjust some notes made throughout the manuscript and presented in this opinion. Finally, it will be necessary to update some information regarding macroeconomic aspects that directly affect the success or inefficiency of the application of predictive tools for decision-making in the waste management and treatment sector.

Additionally, when seeking to achieve sustainable development goals and objectives as described in the collections that make up the 2030 Agenda, it is of crucial importance to permanently analyze the economic viability of any enterprise that seeks to meet ESG requirements and, at the same time, be competitive. Therefore, I ask the authors to mention that the proposed model is an initial attempt to collect and structure information for making technical decisions regarding the reduction of greenhouse gas emissions. If you wish, you can mention that it will be necessary to investigate the best way to make meeting these goals economically viable, especially with the next few years of greater impacts from inflation and gradual slowdown in the global economy as a whole, especially in member countries of the Union. European.

Finally, it is necessary to mention in the title of the manuscript that the survey carried out is a diagnosis/preliminary study.

ABSTRACT:

The summary allows the reader to understand the theme of the manuscript and the research object. However, the information regarding the methodology used was very succinct. The authors must mention the main variables studied to develop the prediction model for decision-making on updating the vehicle fleet for use in different urban solid waste management activities. Additionally, the authors can add the main technical (or financial) restriction that the mathematical model demonstrated as being the main “bottleneck” for fleet renewal and, whether the decision-making model allowed establishing different update horizons.

INTRODUCTION:

In this section the authors seek to present the state-of-the-art in relation to the urban solid waste management system, as well as bringing some concepts about circular economy and environmental education as a way of improving the quality of life of citizens and their responsibilities as a waste-generating entity.

However, when characterizing the “difficulties” encountered by decision makers regarding the establishment of traffic management for collecting vehicles, it is extremely important to mention how much the economic, social and environmental parameters of the population of a location to be served will interfere in choosing the tools and strategies to be applied for more optimized waste management and treatment.

Because, there is a series of vehicles and devices that will be most demanded depending on the nature of the urban solid waste, whether it is a high-standard residential region, commercial region, mixed commercial and residential region, industrial region, etc.

1.1. Fleet transition in solid waste management:

In this section, we spoke very superficially about the energy transition associated with waste management and how these segments are connected through tools and methods based on the principles of circular economy. Additionally, when dealing with the energy transition established by European Union member countries, it only dealt with the need to update fleets.

However, it is necessary to mention how these targets should be met, or revised, taking into account that the vast majority of countries in the world are heading towards increasing inflation and the risk to food and energy security resulting from the disruption of value chains due to the pandemic and the occurrence of wars and conflicts with countries that participate as important stakeholders in maintaining the global supply supply.

At the same time, it is essential to mention whether the objectives for the energy transition are in progress considering the options for generating energy from renewable sources (ecofriendly) to the detriment of the use of coal-fired thermoelectric plants for the European community, or, if you prefer, as the transition of the energy matrix is taking place in Poland. Because, without this information, what may appear to those who read the manuscript is that the objectives for sustainable development will not have achieved their goals effectively, representing a “sum equal to zero” relationship.

MATHERIALS AND METHODS

It is known that, in addition to the development of battery electric vehicles, there are attempts to use engines that process ammonia gas, as well as hybrid combustion engines that allow for greater gains in efficiency by converting the mechanical energy of the axle and bearing system into dynamos and sequencers that make efficiency gains with less carbon gas released.

In this sense, I suggest that the authors make adjustments throughout this section and make existing technological alternatives for replacing the vehicle fleet more prominent. It is not possible to immediately understand that the zero-carbon emission vehicle that the authors mention is the hydrogen cell electric vehicle.

MARKAL – an overwiev

- In addition to allowing prediction regarding the effects of technological obsolescence and effective costs, is it possible to apply the effect of taxation and inflation on the total operating cost?

If so, I suggest citing some technical and/or scientific works that demonstrate the versatility of this predictive methodology.

As a professional and consultant in this segment, I say that the greater complexity in establishing decision-making models with good predictive capacity is limited to the use of a lot of software that restricts the joint compilation of variables to be estimated and, consequently, leads to models with little adjustment.

Fleet Transition Problem

- When analyzing the input parameters and functions associated with the prediction model, it is not possible to verify the effect that this transition would cause in terms of the new costs that a fleet of more environmentally friendly vehicles would impact on the continuity of the action plan established for the partial exchange and periodic analysis of the fleet as a whole. In this sense, I ask that the authors point out whether the proposed model can estimate the effects caused by the transition in terms of new energy and fleet maintenance costs, mainly the costs involving the maintenance and/or replacement of these electric motors, as well as the possible costs of treatment and final disposal.

Add the term “RESULTS AND DISCUSSION” before line 219

Computational Experiments

- Table 2: what do these “unit costs” represent in relation to the current currency? Furthermore, what do emission values represent? Was it a normalization based on some technical information?

- Figure 1: what is the time interval between the periods shown in the figure? How many years (or months) will the fleet be renewed?

- Table 3: in the same way as requested for figure 1 and table 2, authors must mention how “cost of modernization” was assigned and whether they represent codified and/or standardized information.

- Table 4: check the title of this table. The title copies the information from figure 3 and does not faithfully represent the information in the table.

CONCLUSIONS

- Lines 275 – 281: I only consider this paragraph to be the conclusions that the study allows us to make.

The discussions presented below can be rewritten at the end of the previous section. Mainly, due to the fact that it responds to part of the requests I suggested to the authors to improve understanding and reading fluidity.

Author Response

(The authors gave the same response as above.)

Round 2

Reviewer 3 Report

Comments and Suggestions for Authors

Dear Authors,

After corrections and adjustments, I was able to better understand the main idea of the work carried out and reported in this manuscript. It is possible to better understand the state-of-the-art of the study and its perspectives as new data is obtained.

In this sense, I consider the manuscript suitable for publication in Sustainability.

Good job,

Reviewer